# Pectin Remodeling and Involvement of AtPME3 in the Parasitic Plant–Plant Interaction, *Phelipanche ramosa*–*Arabidospis thaliana*

**DOI:** 10.3390/plants13152168

**Published:** 2024-08-05

**Authors:** Cyril Grandjean, Christophe Veronesi, Christine Rusterucci, Charlotte Gautier, Yannis Maillot, Maïté Leschevin, Françoise Fournet, Jan Drouaud, Paulo Marcelo, Luciane Zabijak, Philippe Delavault, Philippe Simier, Sophie Bouton, Karine Pageau

**Affiliations:** 1UMR INRAE 1158 BioEcoAgro, BIOlogie des Plantes et Innovation, Université de Picardie Jules Verne, F-80000 Amiens, France; cyril.grandjean@u-paris.fr (C.G.); christine.rusterucci@u-picardie.fr (C.R.); charlotte.gautier@u-picardie.fr (C.G.); y.maillot@u-picardie.fr (Y.M.); maite.leschevin@u-picardie.fr (M.L.); francoise.fournet@u-picardie.fr (F.F.); 2CNRS, US2B, UMR 6286, Nantes Université, F-44000 Nantes, France; christophe.veronesi@univ-nantes.fr (C.V.); philippe.delavault@univ-nantes.fr (P.D.); philippe.simier@univ-nantes.fr (P.S.); 3Centre Régional de Ressources en Biologie Moléculaire UPJV, Bâtiment Serres-Transfert Rue Dallery—UFR des Sciences, Passage du Sourire d’Avril, F-80039 Amiens, France; jan.drouaud@inrae.fr; 4Plateforme d’Ingénierie Cellulaire & Analyses des Protéines ICAP, Université de Picardie Jules Verne, F-80000 Amiens, France; paulo.marcelo@u-picardie.fr (P.M.); luciane.zabijak@u-picardie.fr (L.Z.)

**Keywords:** cell wall, pectin acetyl esterase, pectin methyl esterase, pectin remodeling enzymes, parasitic weed

## Abstract

*Phelipanche ramosa* is a root parasitic plant fully dependent on host plants for nutrition and development. Upon germination, the parasitic seedling develops inside the infected roots a specific organ, the haustorium, thanks to the cell wall-degrading enzymes of haustorial intrusive cells, and induces modifications in the host’s cell walls. The model plant *Arabidopsis thaliana* is susceptible to *P. ramosa*; thus, mutants in cell wall metabolism, particularly those involved in pectin remodeling, like *Atpme3-1*, are of interest in studying the involvement of cell wall-degrading enzymes in the establishment of plant–plant interactions. Host–parasite co-cultures in mini-rhizotron systems revealed that parasite attachments are twice as numerous and tubercle growth is quicker on *Atpme3-1* roots than on WT roots. Compared to WT, the increased susceptibility in *AtPME3-1* is associated with reduced PME activity in the roots and a lower degree of pectin methylesterification at the host–parasite interface, as detected immunohistochemically in infected roots. In addition, both WT and *Atpme3-1* roots responded to infestation by modulating the expression of PAE- and PME-encoding genes, as well as related global enzyme activities in the roots before and after parasite attachment. However, these modulations differed between WT and *Atpme3-1*, which may contribute to different pectin remodeling in the roots and contrasting susceptibility to *P. ramosa*. With this integrative study, we aim to define a model of cell wall response to this specific biotic stress and indicate, for the first time, the role of PME3 in this parasitic plant–plant interaction.

## 1. Introduction

Among *Orobanchaceae* plants, the branched broomrape, *Phelipanche ramosa* L. Pomel, is an obligate parasitic plant with a weedy life in cropping areas. Its expansion is not controlled to date, and so it is a significant pest in Solanaceae (tomato and tobacco), Brassicaceae (cabbage and rapeseed), and Cucurbitaceae fields, especially in Central Europe [1,2].

*P. ramosa*’s seed germination is achieved by eliciting molecules from host roots, primarily strigolactones. Other host-derived molecules, notably cytokinins, induce the differentiation of the primary root tip into a specialized organ called the haustorium for the establishment of physical and physiological interactions with the host plant [3]. In a compatible interaction, haustorial intrusive cells invade the host root cortex, reach the stele, and successfully connect to the xylem and phloem tissues (similar to a graft junction) (development stage 1, Figure 1). Water and nutrient spoliation from the host results in haustorium growth, which distends host root tissues and forms a storage organ called the tubercle outside of the host root (development stages 2 and 3, Figure 1). A floral meristem develops into a subterranean shoot, emerges above the soil, branches, and blooms. Self-pollination triggers the production of several thousand seeds that can remain viable in the soil for more than twenty years, leading to severe soil pollution [4].

Numerous studies have investigated the dynamics of the plant cell wall in response to abiotic and biotic stresses [5]. The plant cell wall is a complex structure composed of cellulose microfibrils and non-cellulosic neutral polysaccharides embedded in a physiologically active pectin matrix, cross-linking with structural proteins and lignin, depending on the tissue or organ [6,7]. The primary cell wall of growing cells is distinct from the secondary cell wall, which is deposited inside the primary wall of specific cell types with specialized functions. Additionally, the middle lamella, a pectin layer, fills the space between the adjacent cells and firmly adheres to them [8]. In the context of plant parasitism, cell wall-degrading enzymes from haustorial intrusive cells modify the adjacent host cell wall [9]. Many studies in *Orobanchaceae* have focused on pectin remodeling enzymes (PREs), including pectin acetyl esterase (PAE), pectin methyl esterase (PME), polygalacturonases, and pectate lyases [10,11,12,13]. PREs act in concert to weaken the host cell wall, facilitating the progression of haustorial intrusive cells into infested roots. Immunolabeling experiments have revealed the presence of highly de-esterified pectins in host cell walls, correlated with the presence of PMEs and high pectinolytic activity in intrusive cells and the adjacent apoplast [14,15]. Accordingly, high pectinolytic activities from the infecting parasite correlate with high aggressiveness against the host plant [16].

The alteration of cell wall integrity in infected plants serves as a signal that activates effective defensive responses [17]. Cell wall residues such as oligogalacturonides induce basal plant defenses [18]. Oligogalacturonides correspond to oligomers of galacturonic acids released from homogalacturonan, a major cell wall pectin component [19]. Their recognition through wall-associated kinases [20], which are membrane-localized receptors, is considered a system for monitoring pectin integrity that induces a set of defense responses, such as the accumulation of reactive oxygen species and pathogenesis-related proteins [18,21]. Interestingly, a cell wall kinase is overexpressed early in tomato roots challenged by *P. ramosa* [22], likely acting as a sensor of cell wall alterations during broomrape attack.

Among the PME isoforms in *A. thaliana*, PME3 triggers susceptibility to necrotrophic fungal pathogens and parasitic nematodes [23,24]. The responsiveness to oligogalacturonides through the activation of wall-associated kinase receptors increases in the *Atpme3-1* mutant. Consequently, the mutant exhibits low PME activity and a high degree of homogalacturonan methyl esterification in the roots and hypocotyls compared to WT [20,25]. As reported by Pérez-De-Luque et al., in the incompatible interaction between *Vicia sativa* and *Orobanche crenata*, the production of mucilage rich in de-esterified pectin in the xylem vessels of attacked plants triggers incompatibility by obstructing the xylem vessels, thereby preventing the development of the attached parasite [26]. Altogether, the studies carried out to date suggest that pectin and PREs play a significant role in parasitic plant–plant interactions by conditioning both haustorium development and the activation of defense mechanisms in infected roots.

In the present study, we developed an integrative approach using WT and the *Atpem3-1* mutant to capture pectin modifications in *A. thaliana* challenged by the parasitic plant *P. ramosa* and to address the question about the involvement of *AtPME3* in this parasitic plant–plant interaction.

## 2. Results

### 2.1. P. ramosa Has Significantly Fewer Putative PRE-Encoding Genes than A. thaliana

In several plant species, PREs belong to large multigenic families [27]. In *A. thaliana*, 243 PREs have been identified, divided into 12 PAEs, 66 PME, 71 PME inhibitors (PMEI), 26 pectate lyases (PL), and 68 polygalacturonases (PG) [2]; cellwall.genomics.purdue.edu, accessed on 10 May 2021) (Figure 2). According to InterProScan software (https://www.ebi.ac.uk/InterProScan, accessed on 11 May 2021), UniProt sequence data (https//www.uniprot.org, accessed on 15 May 2021), and PREs gene sequences obtained by de novo transcriptome assembly [28], PREs were globally twice less abundant (128) in *P. ramosa* than in *A. thaliana* (243), regardless of the PREs tested, except for the PAE family. Fourteen PAEs were accounted for in the parasite compared to 12 PAEs in *A. thaliana*. Moreover, the number of PREs in *P. ramosa* was closer to that in commelinid monocotyledons [29], which had a type-II cell wall with a significant reduction in pectins and xyloglucans in comparison with the type-I cell wall of *A. thaliana* [30], (https//www.uniprot.org, accessed on 15 May 2021).

### 2.2. Atpme3-1 Is More Susceptible to P. ramosa than WT

No parasite attachment was detected at 6 h after infestation (hai), 12 hai, and 8 days after infestation (dai). The first attachments to WT and *Atpme3-1* occurred during the second week following infestation. At 14 dai, parasite attachments (stages 1 and 2; Figure 1) were significantly more numerous on *Atpme3-1* (median: 11.5) than on WT (median: 4) (Figure 3). Very few tubercles reached stage 3 (Figure 1) at this early time point of infestation (Figure 3C). Later, at 28 dai, the total number of parasite attachments doubled on both WT and *Atpme3-1,* and most of them reached stages 2 and 3. Attachments remained more numerous on *Atpme3-1* (median: 40) than on WT (median: 26). During the following two weeks, the total number of parasite attachments did not change significantly, marking the maximum level of infestation at 28 dai for WT and *Atpme3-1*. However, the parasite continued to develop more rapidly on *Atpme3-1*. At 42 dai, there were no more stage 1 attachments to WT and *Atpme3-1*. The number of stage 2 attachments decreased while the number of tubercles at stage 3 increased, more obviously on *Atpme3-1*.

### 2.3. Atpme3-1 Displays Lower Homoglalacturonan Methyesterification Degree than WT at the Host–Parasite Interface

Immunolabeling experiments were conducted on young control and infected lateral roots to clarify homogalacturonan distribution patterns in WT and *Atpme3-1*, particularly at the host–parasite interface during haustorium maturation (stage-1 attachments at 14 dai). Primary monoclonal antibodies LM19 and LM20 were used to stain low and high methylesterified homogalacturonans, respectively. At this stage, pectin remodeling in host roots could be very intensive due to haustorium-secreted cell wall-modifying enzymes [12,31].

Under control conditions, immunolabeling with the LM20 antibody resulted in a significant signal (green) in the conductive xylem vessels of both WT and *Atpme3-1* roots (Figure 4A,B,E,F), indicating the presence of highly methylesterified homogalacturonans. No labeling was detected using the LM19 antibody, which targets lowly methylesterified homogalacturonans (Figure 4C,D,G,H).

In infected WT roots, weak LM19 and strong LM20 signals were detected at the host–parasite interface (Figure 4K,O and Figure 4I,M, respectively). In contrast, both LM19 and LM20 signals were low in infected *Atpme3-1* roots, indicating lower levels of homogalacturonan methylesterification at the host–parasite interface compared to WT (Figure 4L,P and Figure 4J,N respectively).

### 2.4. The Infestation Modulates the Expression of PREs Genes in a Different Way in WT and Atpme3-1 Both before and after Parasite Attachment

The expression of the PRE gene in host roots under control and infestation conditions was assessed by using primers specific to *A. thaliana* sequences (Appendix A). The expression patterns of PME and PAE gene families in roots of WT and *Atpme3-1* were measured at three time points: pre-attachment (6 h and 12 h) and post-attachment (14 days) of *P. ramosa* (Figure 5A–C, respectively).

Among the 12 PAE-encoding genes in *A. thaliana*, only *PAE7* presented a strong expression, which was 1000 times higher in comparison with the other *PAE* genes (Appendix A). Under control conditions, WT and *Atpme3-1* displayed similar levels of *PAE7* expression only at the early time point of 6 hai (Figure 5A—PAE). At 12 hai and 14 dai, *PAE7* expression was significantly lower in the mutant compared to WT (Figure 5B,C—PAE). Infestation modulated *PAE7* expression differently in WT and *Atpme3-1* roots, thus dropping at 6 hai and 12 hai in WT but only at 6 hai in *Atpme3-1* (Figure 5A,B—PAE). Later, modulation in the mutant consisted of *PAE7* overexpression at 6 hai and 14 dai, while decreasing and unchanged levels were observed in the infested WT at 6 hai and 14 dai, respectively (Figure 5A,C—PAE). Finally, WT displayed a higher level of *PAE7* expression than *Atpme3-1* at 14 dai in both control and infestation conditions (Figure 5C—PAE).

Among the 66 gene members of the PME family in *A. thaliana*, only 8 genes were highly expressed at 6 hai in both WT and *Atpme3-1*. *PME18* and *PME31* were consistently the most expressed genes (Figure 5A—PME-WT and A—PME-*Atpme3-1*). Five expressed genes were common to both genotypes: *PME17*, *PME18*, *PME31*, *PME35,* and *PME51*. *PME40* and *PME62* were expressed particularly in WT, whereas *PME41* was expressed only in *Atpme3-1*.

Comparatively, fewer *PME* genes were highly expressed in both genotypes at 12 hai, including *PME18*, *PME17*, *PME31*, *PME35,* and *PME51* in WT, and only *PME18* and *PME62* in *Atpme3-1* (Figure 5B—PME-WT and B-PME-*Atpme3-1*). The expression of these genes also decreased at this time point in response to infestation (Figure 5B—PME-WT), except for *PME 18* expression, which highly increased in *Atpme3-1* (Figure 5B—PME-*Atpme3-1*).

The expression of several *PME* genes, especially *PME 40*, enhanced at 14 dai in the infected WT (Figure 5C—PME-WT). Along with *PME18* and *PME31*, *PME40* became one of the most highly expressed *PME* genes in the infected WT. A similar pattern was observed in infected *Atpme3-1*, particularly for *PME62* and *PME31* (Figure 5C—PME- *Atpme3-1*).

More generally, the expression of certain *PME* genes increased in response to infestation in both genotypes. However, this response was observed much earlier, as soon as 12 hai, in *Atpme3-1* (Figure 5B—PME-WT and -*Atpme3-1*).

### 2.5. The Infestation Modulates PME and PAE Activities in WT and Atpme3-1 before and after Parasite Attachment

Due to the presence of the haustorium, it was impossible to mechanically isolate the parasitic tissues from the infected host roots. As mentioned previously, WT and *Atpme3-1* roots were not infected before the second week of infestation (Figure 3). Thus, roots were free of parasite attachments at 6 hai, 12 hai, and 8 dai. In contrast, host roots were infected at 14 dai with very young parasite attachments (development stages 1 and 2). However, their number was relatively low (Figure 3), and their total weight was also extremely low compared to the weight of the host root. PME and PAE activities measured from infected roots at 14 dai could be considered to belong to the host (Figure 6).

At the early time point corresponding to 6 hai, WT and *Atpme3-1* showed similar PAE activities under control conditions, while *Atpme3-1* exhibited lower PME activity compared to WT (Figure 6B). Under infestation, PAE and PME activities significantly decreased in both genotypes, consistent with the decreased expression of *PAE7* and all *PME* genes at this time point (Figure 5).

At the time point corresponding to 12 hai, PAE and PME activities in control conditions were significantly higher in WT than in *Atpme3-1* (Figure 6A,B). PAE activity in the control WT increased when compared to 6 hai, while PME activity remained relatively stable. Conversely, the control *Atpme3-1* exhibited a slight increase in PME activity at 12 hai. Under infestation, a slight decrease in PAE activity was observed in WT, with no significant change in *Atpme3-1*. On the contrary, *Atpme3-1* exhibited no change in PAE activity and a significant decrease in PME activity at 12 hai in response to infestation. In this way, infestation-induced changes in PAE activity and *PAE7* gene expression matched at 6 hai and 12 hai in WT and only at 6 hai in *Atpme3-1* (Figure 5A—PAE and Figure 6A). Moreover, this finding discords with the decrease in PME activity in *Atpme3-1* at 12 hai (Figure 5B—PME-*Atpme3-1*, and Figure 6B).

Later, at 8 dai, the infestation did not impact PAE activity in both genotypes and PME activity in WT (Figure 6A,B). Only PME activity slightly decreased in *Atpme3-1* upon infection.

At 14 dai, PME activity was unaffected by infestation in both genotypes (Figure 6B), whereas PAE activity strongly decreased in the infected WT and slightly increased in infected *Atpme3-1* (Figure 6A,B).

Overall, PME activities in roots were almost twice as low in *Atpme3-1* compared to WT under both control and infestation conditions and at all the time points (Figure 6B).

## 3. Discussion

Pectins are major components of the primary plant cell wall and ensure cohesion between cells [32]. Pectin remodeling occurs during plant growth and development due to various PRE enzymes, including PAE and PME [28]. Maintaining cell wall integrity is crucial in the adaptation and establishment of tolerance mechanisms for stress, particularly biotic stress [5]. Mechanisms controlling this integrity have to be investigated, particularly in the context of parasitic plant–plant interactions, for which information is scarce.

Guénin et al. carried out the first investigations on pectin remodeling in roots of the mutant *Atpme3-1* [25]. In addition to a low HG content, the mutation induced a significant decrease in PME activity in accordance with a modified pectin pattern in favor of highly methylesterified homogalacturonans. Given that such changes normally limit the action of hydrolytic enzymes of pathogens [33,34], *Atpme3-1* is effectively less susceptible to pathogenic microorganisms and nematodes [9,24], revealing the contribution of AtPME3 in susceptibility to those pathogens. Moreover, in the interaction between *A. thaliana* and the nematode *Heterodera schachtii*, Hewezi et al. showed that AtPME3 binds to the effector, the Cellulose-Binding Protein, leading to changes in the host cell wall that facilitate infestation [23]. The present study confirms the limitation in PME activity in *Atpme3-1* roots (Figure 6B,C). However, *Atpme3-1* turns out to be much more susceptible to the parasitic plant *P. ramosa* than WT, and the parasitic plant develops more rapidly on *Atpme3-1* roots (Figure 3). These findings indicate that pectin remodeling in *Atpme3-1* roots promotes successful parasite attachment and tubercle development and, finally, that changes in cell walls induced by *PME3* mutation affect the susceptibility of A. thaliana to pathogens differently according to the infecting organism.

In the present study, immunohistochemical studies using LM19 and LM20 antibodies revealed highly methylesterified pectins in the vascular cell walls in WT and *Atpme3-1* roots under control conditions (Figure 4A,B). According to Guénin et al., *Atpme3-1* displayed a methylesterification degree of uronic acids 1.4 times higher compared with WT in 10 d-old roots and hypocotyls (FT-IR technology, [25]). Such a difference could not be detected at tissue and cellular levels in the present study. On the other hand, immunohistochemical studies on infected roots harvested early during infection (stage 1-attachments, 14 dai, Figure 2) emphasized a decrease in pectin methylesterification at the host–parasite interface in *Atpme3-1* in response to infection (Figure 4B,F). In contrast, the infection did not induce changes in pectin methylesterification in WT roots (Figure 4A,E). This finding suggests that enhanced susceptibility in *Atpme3-1* may be attributed to enhanced cell wall release at the host–parasite interface, which facilitates haustorium development and parasite attachment. It thus reinforces the interest in assessing the expression of PRE-encoding genes and the associated enzyme activities in WT and *Atpme3-1* roots during infestation.

Pectin esterases, including PAE and PME, are essential for pectin remodeling [35,36]. Vieira Dos Santos et al. reported that infestation triggers general signaling pathways involved in plant defense before parasite attachment to *A. thaliana* (WS) roots [37]. Our results show that the host roots perceive the parasite early during infestation in both WT and *Atpme3-1*, resulting in a concomitant reduction in PAE and PME gene expression and activities at 6 hai, well before the parasite penetrated the host roots (Figure 5A–PAE and Figure 6A). Among the PAE multigenic family, only *PAE7* showed decreased expression in both infested genotypes. In addition, among the 66 PME-encoding genes, only 8 were expressed in WT and *Atpme3-1* and also exhibited decreased expression early at 6 hai in response to infestation (Figure 5A–PME). Four of these genes—*PME 17, PME 18, PME 31,* and *PME 35*—are strongly expressed in response to pathogens, in particular bacteria and nematodes [24,28,38,39]. Conversely, WT and *Atpme3-1* responded differently to infestation at 12 hai. For example, the PME-encoding gene and *PAE7* expression were still affected by infestation in WT, while *PME18* and *PAE7* overexpressed in *Atpme3-1* (Figure 5A,B). Changes in gene expression and enzyme activities did not match at this time point of infestation since PAE activity declined in WT but not in *Atpme3-1,* whereas PME activity declined in *Atpme3-1* but not in WT (Figure 6A). Such mismatches were also found for PME at 14 dai and also in previous studies [40,41]. They notably address the question about the involvement of PME inhibitors in regulating PME activity [42,43], notably given that five PME of WT and *Atpme3-1* display an N-terminal extension (PRO region) with similarities with the PME inhibitor domain, Pfam04043 [44]. Later, at 14 dai, enzyme activities tended to be less affected in the roots of both genotypes, except for PAE activity in WT, which decreased strongly (Figure 6B). Randoux et al. suggested that the degree of pectin acetylation is a key point in the response of wheat to mildew since treatment with acetylated oligoglacaturonides prior to infection inhibited the growth of the pathogenic haustorium [45]. Our findings thus suggest that WT might maintain a higher degree of pectin acetylation in roots, limiting parasite attachment by preventing the action of potential parasite’s polygalacturonases, resulting in lower susceptibility to *P. ramosa* in comparison to *Atpme3-1*.

Lowly methylesterified pectins were detected at the host–parasite interface in the infected *AtPME3* mutant, while PME activity decreased (Figure 3 and Figure 6). These findings show that global changes in PME activity in infected roots of the mutant do not correlate with local-specific modification in pectin methylesterification. These results suggest that *AtPME3* mutation induces changes in the *A. thaliana* cell wall in favor of pectin demethylesterification via the parasite’s PREs at the host–parasite interface, resulting in cell wall release and promoting parasite invasion. Further studies are required at tissue and cell levels in both host and parasite to fully understand these plant–plant interactions. However, unlike in facultative hemiparasitic plants, genetic transformation, and thus mutants, is currently unavailable for holoparasitic plants like *P. ramosa*. This prevents genetic approaches from investigating the role of parasite PREs in infection, as has been achieved in the facultative parasite *P. japonicum*, where changes in *PjPME* and *PjPMEI* expression were associated with tissue-specific modification in pectin methylesterification during haustorium development [46].

In addition, understanding the role of PREs in the parasitic plant–plant interaction is challenging due to the fact that the infecting organism is also an angiosperm, which makes it more challenging for host plants to recognize it as a pathogen, and that the plant cell wall may actually appear as the assembly of multiple specific cell wall microdomains. Homogalacturonans vary in size with various degrees of polymerization and charge [47]. Moreover, the multigenic families of PREs are similar in size (about 70 genes each in *A. thaliana* [48,49]), rendering theoretically plausible the combinatory interactions of individual members. Additionally, the precise and dynamic modulation of extracellular pH controls PRE activities, and in particular PME and polygalacturonases [50]. Complete functional studies, including host and parasite PRE, should be addressed in the future within parasitic plant–plant interaction, particularly when it comes to understanding the molecular interactions between various cell wall components.

## 4. Materials and Methods

### 4.1. Plant Material and Growth Conditions

*Phelipanche ramosa* (L.) Pomel (genetic type 1, lab reference: Pram10) seeds were collected in 2011 from mature broomrape flowering spikes in an oilseed rape field at Saint Martin de Fraigneau (France) and stored at 25 °C in the dark before use. *Arabidopsis thaliana* (L.) Heynh WS (Wassilewskija) ecotype and WS *Atpme3-1* mutant (isolated from the Versailles T-DNA insertion collection (FLAG585E02)) were used for co-cultivation experiments in Petri dishes. Two hundred *A. thaliana* seeds of each genotype were surface-sterilized by placing them in an Eppendorf tube containing 70% (*v*/*v*) ethanol and 0.05% (*v*/*v*) SDS up to 2 mL. The tube was placed on a stirring table for 5 min (70 rpm), and then the liquid was removed and replaced with 90% (*v*/*v*) ethanol. The tube was stirred for 5 min again, the liquid was removed, and seeds were put to dry overnight under a suction hood. Using a sterile toothpick, seeds were placed on square Petri dishes (12 cm × 12 cm) containing ½ MS MES medium and incubated at 21 °C in a growth chamber (16 h light, 120 µmoles PAR m^−2^ s^−1^, 8 h dark) for 21 d.

### 4.2. Induction of Broomrape Seed Germination

*P. ramosa* seeds (200 mg) were surface-sterilized for 5 min with 12% (*v*/*v*) sodium hypochlorite in a 50 mL plastic tube and thoroughly rinsed three times with sterile distilled water. Seeds were then suspended (10 mg mL^–1^) in 1 mM of HEPES, pH 7.5, 0.1% (*w*/*v*) PPM Plant Preservative Mixture and incubated for 7 d at 21 °C in the dark to be conditioned. Then, germination was induced by adding *Rac*-GR24, a synthetic germination stimulant (final concentration: 10^−9^ M in 0.2% acetone *v*/*v*). Subsequently, seeds were incubated for 3 d at 21 °C in the dark for germination. Seeds, considered as germinated when the radicle protruded out of the seed coat, were used for plant infestation.

### 4.3. Co-Cultivation Experiments

Twenty-one-day-old *A. thaliana* seedlings (WT or *Atpme3-1*) were transferred onto filter paper and placed in cut square plates (120 × 120 × 17 mm, Greiner, France) containing a uniform layer of rockwool moisturized with 50 mL of ½ TT medium [51]. Each plate contained 5 plantlets. Plates were sealed and incubated vertically at 21 °C in a growth chamber (16 H light, 120 µmoles PAR m^−2^ s^−1^, 8 h dark, 70% humidity) for 7 d and supplied every 3 d with 10 mL of ½ TT medium. After 7 d, the infestation was induced by covering *A. thaliana* roots with 2 mL of germinated *P. ramosa* seeds (4000 germinated seeds per plate; germination rate: 86.03 ± 4.07%). Previously, the seeds were rinsed three times with distilled water to remove any trace of GR24 and its solvent (0.2% acetone *v*/*v*) before being used. Plates were supplied every 3 d with 10 mL of ½ TT medium during 49 d. Controls consisted of non-infested plants.

Susceptibility to *P. ramosa* was assessed by counting parasite attachments to the host roots under a stereo microscope (Olympus SZX10, Olympus Europa GmbH Hamburg, Germany) at early time points (6 h after infestation (hai), 12 hai and later (8 days after infestation (8 dai), 14 dai, 28 dai, and 42 dai), and by distinguishing 3 developmental stages (Figure 1). Each modality (control and infested WT and *Atpme3-1* plants) was performed using at least 5 plates, with each plate containing 5 plants. Data are means ± confident intervals (n ≥ 25, multiple *t*-Tests (FDR correction 0.1%)).

The roots of control and infested plants were also collected at 6 hai, 12 hai, 8 dai, and 14 dai and immediately placed in liquid nitrogen for further molecular analyses and enzymatic assays.

### 4.4. Bioinformatic Analyses

PRE families (PME, PAE, PME inhibitors, polygalacturonases, and pectate lyases) are associated with specific Pfam domains (PME: PF1095, PME inhibitors: PF04043, PAE:PF03283, polygalacturonases: PF00295, pectate lyases: PF00544, https//www.uniprot.org). Using InterProScan software (https://www.ebi.ac.uk/InterProScan (accessed on 20 March 2021)) and UniProt sequence data (https//www.uniprot.org), Pfam domains were investigated in PRE protein sequences and *PRE* gene sequences in *P. ramosa* obtained via de novo transcriptome assembly [29,52].

### 4.5. Cytological Analyses

Control and infected secondary lateral roots harvested at 14 dai were cut into 1 cm and 0.5 cm from each side of the parasite attachment (stage 1, Figure 1) area, respectively. Root segments were immediately fixed in 4% (*w*/*v*) paraformaldehyde dissolved in 0.1 M of sodium phosphate buffer (pH 7.4), to which 1% (*w*/*v*) sucrose and 0.05% (*v*/*v*) tween 20 were added [53]. Fixation of root segments was performed by successive infiltration steps of 15 min under 25 kPa at room temperature. Paraformaldehyde was then removed from samples via successive ethanol baths from 30 to 90% for 30 min under shaking conditions, followed by 2 h in 100% ethanol. Root samples were then included by carefully placing them in 100% EtOH/LR White (*v*/*v*) for 5 h followed by pure medium-grade acrylic LR White resin (Agar Scientific, http://www.agarscientific.com) for 2 days, taking care to replace the resin every day. Samples were finally placed in the center of capsules and placed in a heat chamber for 24 h (67 °C). Polymerized samples were cut using an ultramicrotome (Leica ultracut UCT), and 2 µm sections were collected on poly-L-lysine-treated well glass slides. Plates (n = 5 per modality) were analyzed under bright-field optics with a light microscope (Eclipse 90i, Nikon, Hamamatsu, Japan) and a stereoscopic microscope (SteREO Discovery V20, CARL ZEISS, Jena, Germany) depending on the desired resolution.

Immunolabeling of homogalacturonans from the selected samples was realized according to Turbant et al. [54] using the primary monoclonal antibodies LM19 and LM20, which stain low and high methylesterified homogalacturonans, respectively (PlantProbes, University of Leeds, Leeds, www.plantprobes.net). In addition, calcofluor White was used to visualize cell walls by staining both cellulose and β-1,4 glycans. Samples were imaged with a confocal laser microscope (LSM 780, Carl Zeiss). Images were acquired with a ×40 HCX PL APO CS 1.25 NA oil objective with the following parameters: image dimension of 512 × 512, scanning speed of 400 Hz, line average of 8, and pinhole of 1 airy unit. Laser power and gain settings for each PMT (PhotoMultiplier Tubes) were slightly adjusted individually for each sample. Images were collected in 8-bits per pixel. All recordings were performed at room temperature (20–25 °C). Image processing was performed with Zen imaging software (Zen Black version-Zeiss) and ImageJ (W. Rasband, National Institutes of Health).

### 4.6. Targeted Transcriptomic Analyses

Total RNA was extracted from 100 mg of control and infested roots (6 and 12 hai, 14 dai) ground with liquid nitrogen using Macherey-Nagel™ RNA Plant and Fungi NucleoSpin™ according to the manufacturer’s recommendations. Genomic DNA was removed using Turbo DNA-free^™^ kit (Ambion), according to the manufacturer’s protocol. cDNA synthesis was performed using 4 μg of RNA, 2.5 μM oligo (dT)_18_, and the Transcriptor High Fidelity cDNA Synthesis Kit (ROCHE) using the manufacturer’s protocol. RT-qPCR analyses were performed on 1/20 diluted cDNA. For real-time quantitative PCR, the LightCycler 480 SYBR Green I Master (Roche; catalog No. 04887352001) was used on 384-well plates in the LightCycler480 Real-Time PCR System (Roche). The crossing threshold values for each sample (the number of PCR cycles required for the accumulated fluorescence signal to cross a threshold above the background) were acquired with the LightCycler 480 software (Roche) using the second derivative maximum method. Primers used are specific to the host *A. thaliana* and are shown in Appendix A. Stably expressed reference genes (*APT1, TIP41,* and *CLATHRIN*) were selected using GeNorm software [55] and used as internal controls to calculate the relative expression of target genes according to [56]. *AtPME* and *AtPAE* gene expression was measured using stably expressed reference genes mentioned above in two biological samples and two technical replicates per biological replicate. As the results from the two biological samples showed similar changes in gene expression during infection, only the results obtained from one of the two biological replicates and with *APT1* as internal control are shown.

### 4.7. Global Enzyme Assays

Enriched weakly bound cell wall proteins were extracted from roots (6 hai and 12 hai and 8 dai and 14 dai) according to [57] for PME and PAE enzyme assays. Briefly, 100 mg of frozen root powder was homogenized in 300 µL of 50 mM sodium phosphate buffer (pH = 7.5) containing 2 M of sodium chloride. The homogenate was incubated at 4 °C for 30 min under shaking. After centrifugation and supernatant recovery, a second extraction was carried out on the pellet in the same conditions. The supernatants were mixed and desalted using a citrate-phosphate buffer pH 6.5 (McIlvaine’s buffer) containing 100 mM of sodium chloride. The proteins were quantified with bovine serum albumin as standard [58].

PME activity was measured according to [59]. The protein extract (5 µL) was incubated with 95 µL of 50 mM sodium phosphate buffer (pH 7.5) containing 0.025 U of alcohol oxidase (A2404, Sigma-Aldrich, St. Louis, MO, USA) and 100 µg of 90% methylesterified citrus pectin (P9561, Sigma-Aldrich, St. Louis, MO, USA). After 30 min of incubation at 28 °C, 100 µL of staining solution containing 20 mM of pentane-2,4-dione and 50 mM of glacial acetic acid in a 2 M ammonium acetate buffer were added. Absorbance at 420 nm was measured in a microplate reader (Powerwave, Biotek, Colmar, France) after a 15 min incubation at 68 °C. PME activity was determined with reference to a methanol standard curve and expressed in nmol of methanol min^−1^ µg^−1^ protein.

PAE activity was measured from protein extract using triacetin (525073, Sigma-Aldrich, St. Louis, MO, USA) and sugar beet pectin (42% methylesterification and 31% acetylation degrees, CP Kelco) as substrate, respectively. A total of 100 mM of Triacetin was prepared in McIlvaine’s buffer (pH 6.5) containing 100 mM sodium chloride. Sugar beet pectin (10 mg) was suspended in 1 mL of the same buffer. Activity was measured with 130 µL of substrate and 20 µg of protein extract in a final volume of 150 µL incubated at 40 °C for 2h. The amount of released acetic acid was determined using the Megazyme acetic acid kit (Megazyme, K-ACETRM) at 340 nm using a microplate reader (BioTek PowerWave). Data are mean ± confident intervals (n = 3 biological replicates, *t*-Test).

## Figures and Tables

**Figure 1 plants-13-02168-f001:**
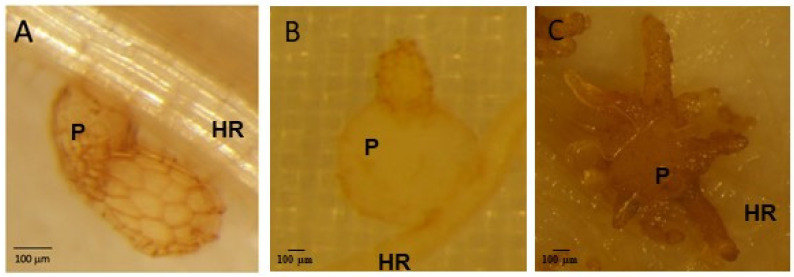
Early development stages of *P. ramosa* attachments to *A. thaliana* roots. (**A**) Stage 1 (parasite invasion into HR, haustorium maturation), (**B**) Stage 2 (mature haustorium, young tubercle), (**C**) Stage 3 (growing tubercle with adventitious roots, “spider” phenotype”). HR: host root, P: parasite.

**Figure 2 plants-13-02168-f002:**
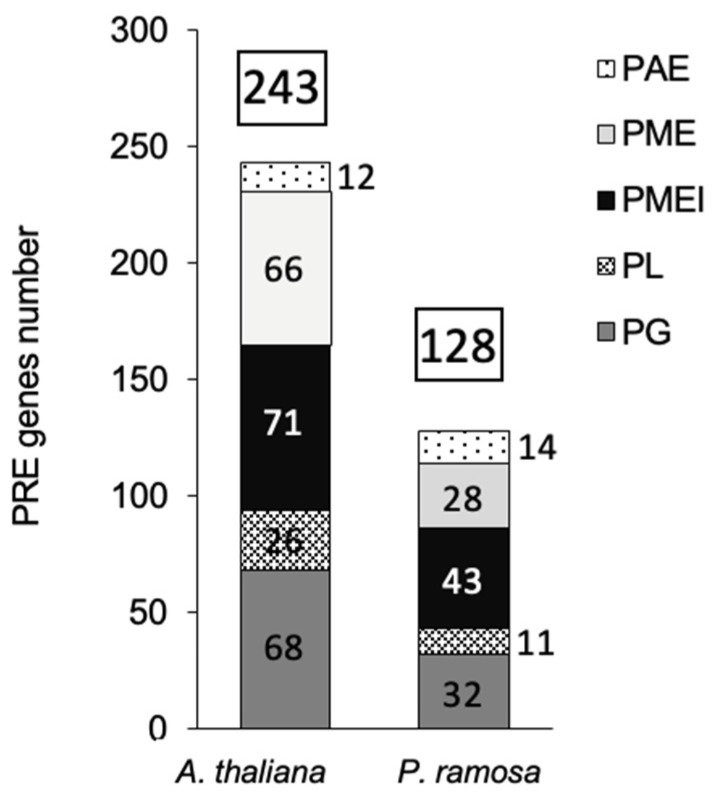
Number of *PREs* putative genes in *A. thaliana* and *P. ramosa***.** PREs families, associated with a specific Pfam domain (PME: PF1095, PMEI: PME inhibitors: PF04043, PAE: PF03283, PG: polygalacturonases: PF00295, PL: pectate lyases: PF00544, https//www.uniprot.org), were found according to InterProScan software (https://www.ebi.ac.uk/InterProScan (accessed on 20 March 2021)), UniProt sequence data (https//www.uniprot.org), and PRE gene sequences obtained by de novo transcriptome assembly [29].

**Figure 3 plants-13-02168-f003:**
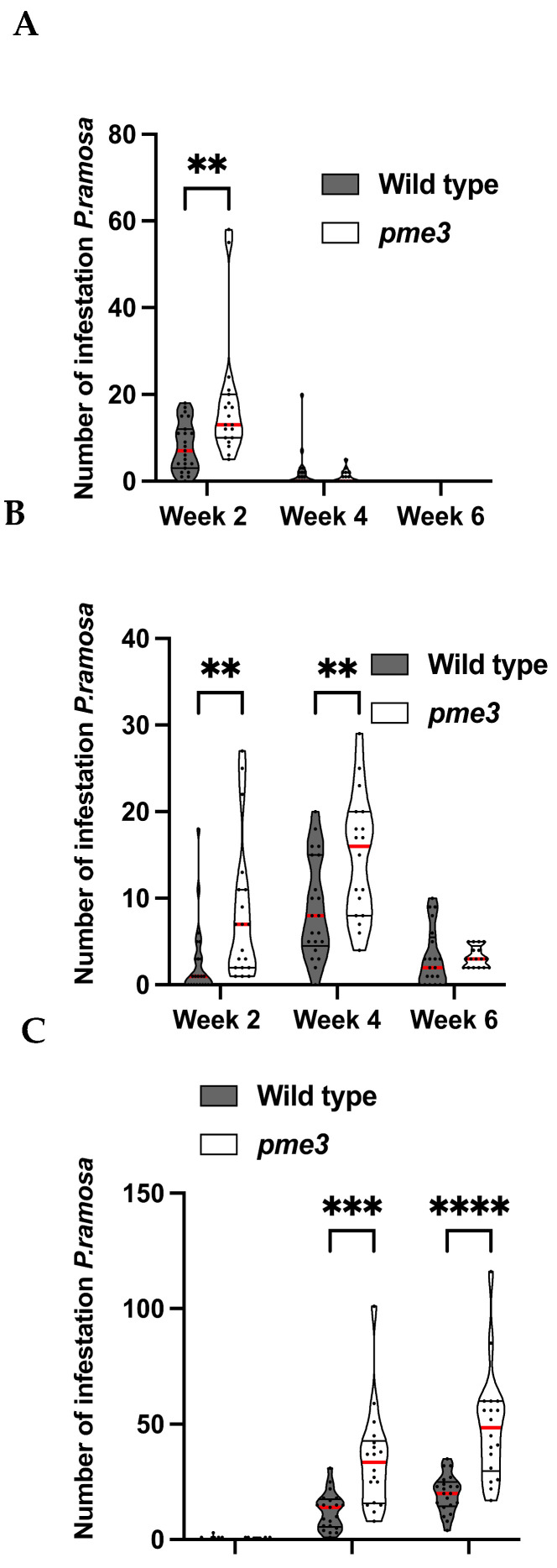
Number of *P. ramosa* attachments on roots of WT or *Atpme3-1* along different development stages (truncated violin plot). (**A**) Stage 1 (haustorium maturation), (**B**) Stage 2 (mature haustorium), (**C**) Stage 3 (growing tubercle with adventitious roots: “spider phenotype”). See Figure 1 for pictures of development stages. Red line represents the median, black line the quartiles, and black dot represents biological replicate (n ≥ 3 biological replicates per genotype). **, *p* < 0.01, ***, *p* < 0.001, ****, *p* < 0.0001, multiple *t*-Test (FDR correction 0.1%).

**Figure 4 plants-13-02168-f004:**
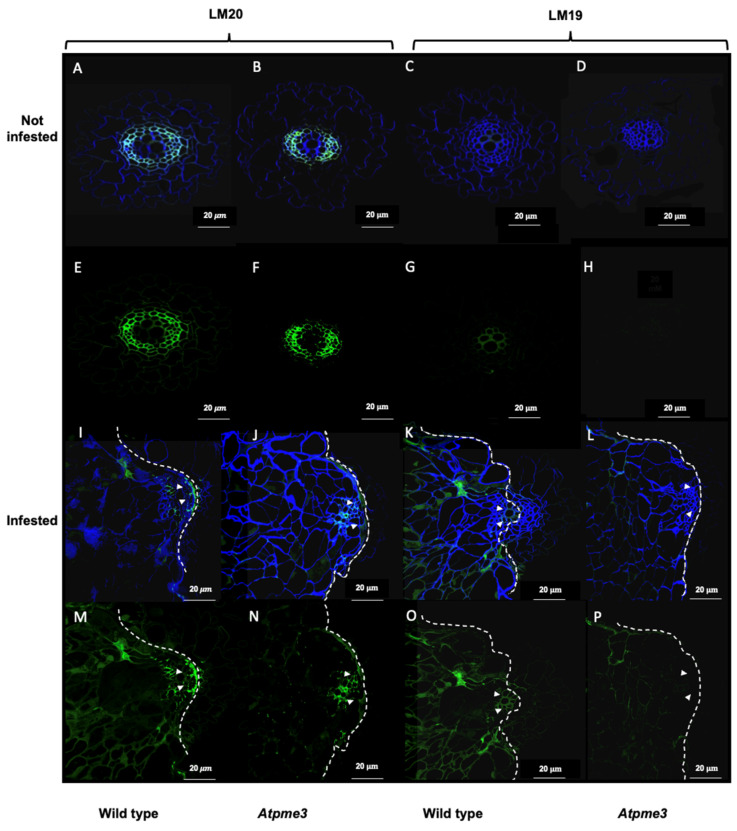
Distribution patterns of homogalacturonans at the host–parasite interface. The parasite is at early stage 1 at 14 dai (Figure 1). Sections of control WT and *Atpme3-1* roots are shown in panels (**A**,**C**,**E**,**G**) and (**B**,**D**,**F**,**H**), respectively. Sections of infected roots from WT and *Atpme3-1* are shown in panels (**I**,**K**,**M**,**O**) and (**J**,**L**,**N**,**P**), respectively. Sections were labeled with LM19-calcofluor (**C**,**D**,**K**,**L**) and LM20-calcofluor (**A**,**B**,**I**,**J**) antibodies, which recognize low and high methylesterified homogalacturonans, respectively. Additionally, sections were labeled with LM19 alone (**G**,**H**,**O**,**P**) and LM20 alone (**F**,**H**,**N**,**P**), respectively (green). Calcofluor White, which stains both cellulose and other b-1,4-glycans, was used to visualize cell walls (blue). White arrows indicate the localization of homogalactoronan pattern modification at the host–parasite interface. The outlines of the host–parasite interface were manually drawn (white dashed lines). Bars = 20 μm.

**Figure 5 plants-13-02168-f005:**
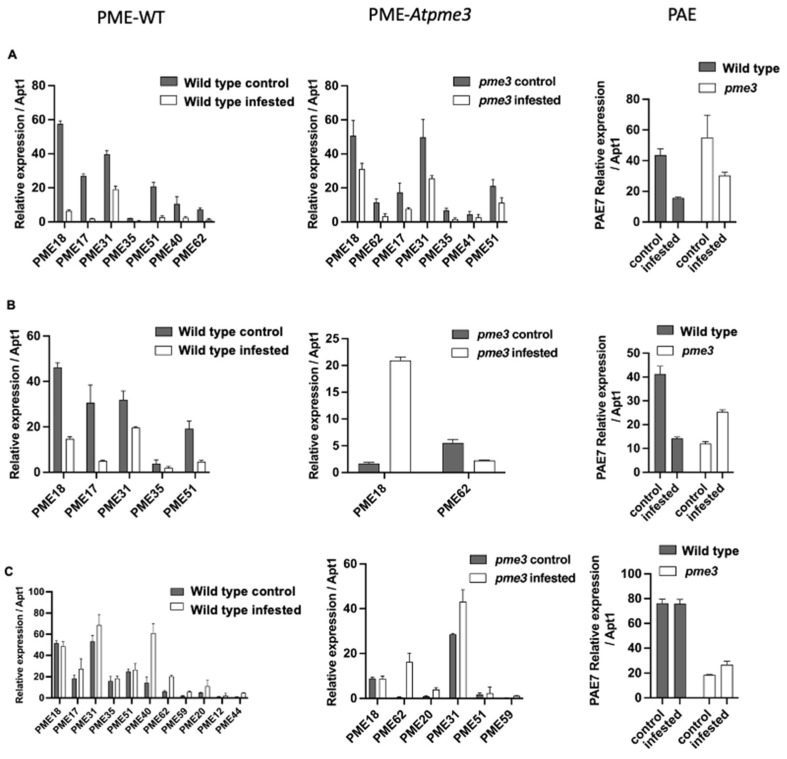
Expression patterns of *PME* and *PAE* gene families in roots of WT and *Atpme3-1* at pre-attachment ((**A**): 6 hai, (**B**): 12 hai) and post-attachment (**C**): 14 dai) of *P. ramosa*. Target genes were normalized to the housekeeping gene *APT1* as internal control (n = 2 technical replicates per genotype and condition).

**Figure 6 plants-13-02168-f006:**
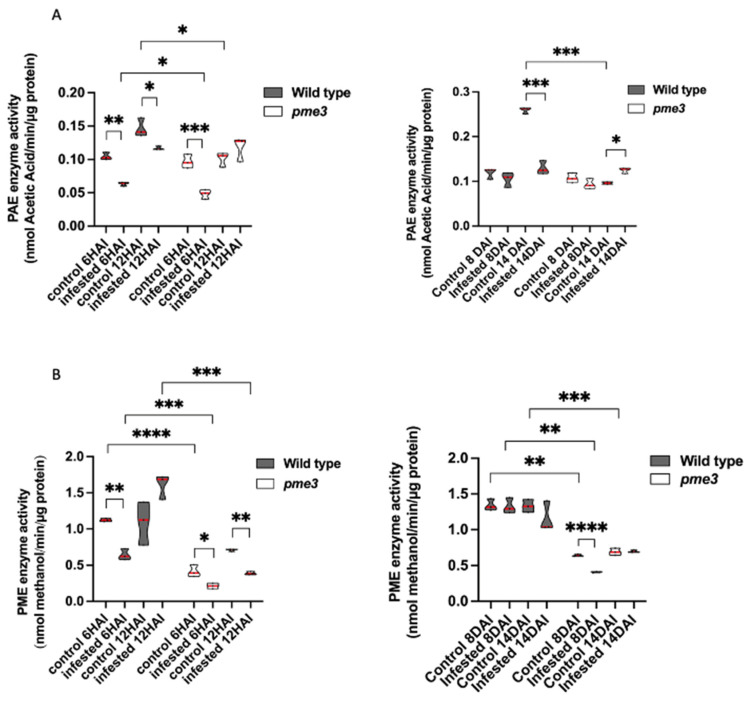
PRE enzymatic activities (PAE activity (**A**) and PME activity (**B**)) in roots of WT and *Atpme3-1* at pre-attachment ((**A**): 6 hai, (**B**): 12 hai) and post-attachment (8 dai, 14 dai) of *P. ramosa*. Red line represents the median, and black line represents the quartiles (n = 3 biological replicates per genotype). *, *p* < 0.05, **, *p* < 0.001, ***, *p* < 0.0001, *****, p* < 0.00001, *t*-Test.

## Data Availability

The raw data supporting the conclusions of this article will be made available by the authors upon request.

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
