# Peer review of "Pectin Remodeling and Involvement of AtPME3 in the Parasitic Plant–Plant Interaction, Phelipanche ramosa–Arabidospis thaliana"

_plants, 2024, doi:10.3390/plants13152168_

Round 1
Reviewer 1 Report
Comments and Suggestions for Authors
Overall, it is a nice piece of work. You may re-think whether figure 6 should be in Methods, or in Results. Figure 2 and 5 seem of poorer quality (pixelated). Could you please improve the quality.
Author Response
We thank reviewers for their careful reading of our manuscript. In the following, we address the points raised that helped to improve the manuscript. Major corrections are highlighted in the revise manuscript.
First, the reviewers consider that a moderate editing of English language is required. Proposed corrections (reviewer 4) are relevant and then improve the revised version. In addition, an experienced English-speaking colleague reviewed the manuscript.
As you encourage us to consider, we reduced the number of references, notably self-citations (13.5% in the revised manuscript, 59 references) versus 19% in the original manuscript, 64 references), to keep only the most relevant citations we consider. We added reference 46 (Leso et al. 2024 in Plant Physiol.) in the revised manuscript (see the highlighted part in Discussion, lines 366-378, see below in cover letter). As mentioned by reviewer 3, we kept the references consistent in the revised text (we used only number in the text, and not name and date). We reduced also number of abbreviations to make reading easier.
We also provided additional information on the fixation of root samples in the section 4.5 cytological analyses (lines 452-459, Materials and Methods).
As proposed by reviewer 1, Figure 6 showing early development stages of the parasitic plant is the first Figure (Figure 1) in the revised version, to make understanding of the results easier. Consequently, we adjusted Figure numeration. We improved also quality of the Figures (notably Fig. 3 and Fig. 6 in the revised manuscript). We redesigned figure 3 horizontally. As required by reviewer 3, legend of figure 3 (in the revised manuscript) was improved.
In addition, reviewer 2 addressed following questions and comments:
- The first one concerns Atpme3-1 mutant, which “is characterized by reduced PME activity in roots while the degree of pectin methylesterification is also decreased there”. The reviewer asks “why PME activity reduction leads to decrease in pectin methylation?”
It’s a relevant question. Global PME activity was measured in A. thaliana roots while high methylesterified pectins were detected locally at the host-parasite interface. Our findings show that global changes in PME activity in mutant’s infected roots do not correlate with changes in pectin methylesterification specifically at the host-parasite interface, and suggest that PME3 mutation induce changes in A. thaliana cell walls that are unfavorable to pectin demethylesterification by parasite’s PREs at the host-parasite interface. A better understanding of the effects of AtPME3mutation in this host-parasite interaction requires further studies at tissue and cellular levels in both host and parasite. Please, see lines 372-378 in the revised manuscript.
- The reviewer also asks why authors do not discuss the activity (or changes in activity) of the Phelipanche ramosa pectin-modifying enzymes and their different activity towards host plant in case of WT or mutant plant?
We could measure only PREs activities in non-infected and newly-infected roots of the host plant (see lines 259-266 in the revised manuscript), but not specifically in the infecting P. ramosa. Unfortunately, genetic transformation and thus mutants of interest are not available in holoparasitic plants such as P. ramosa, due to the obligatory nature of parasitism (in contrast to facultative hemiparasitic plants). This limits possibilities to study specifically the implications of parasite’s PRE in infection or haustorium development, as it has successfully done recently for P. japonicum(Leso et al. 2024 in Plant Physiology). We added this reference to improve Discussion in the revised manuscript. Please see lines 375-378.
- The reviewer suggests that “events that take place in haustoria of a parasite plant, upregulation and downregulation of the PREs (see Leso et al. 2024) which reports modification of pectin methylesterification status in the haustoria of the facultative hemiparasitic plant P. japonicum should be referred to, in addition to the plant cell wall homeostasis regulated by brassinosteroid feedback among other mechanisms should be also discussed in a context of plant-plant interactions”.
We are agree with the reviewer’s comments concerning BR signaling and cell wall homeostasis, but we think that adding this information to the discussion is not fully necessary to discuss our results.
The reviewer 3 addressed following questions and comments:
- "The figure legend for figure 4 needs more explanation. What are the time points for A, B, and C? You mention four time points"
We precise in the text that expression patterns of PME and PAE gene families in roots of WT and Atpme3-1 were measured with primers specific to A. thaliana sequences, at three time points: pre-attachment (6 h and 12 h) and post attachment (14 days) of P. ramosa (Figure 5, A, B and C, respectively). Moreover, no parasite attachment was detected at 6 and 12 hours after infestation. The first attachments to WT and Atpme3-1occurred during the second week following infestation, at 14 days after attachment (Figure 1).
- "Shouldn’t we know the expression levels of PME3 in the wild type, and perhaps to be sure, in the pme3 mutant?"
According to Guénin et al (2011), in wild type, AtPME3 is ubiquitously expressed in A. thaliana, particularly in vascular tissues. In all the organs tested (roots, leavesand stem), PME activity was reduced in the atpme3-1 mutant (a homozygous T-DNA line for AtPME3) compared with the wild type. Moreover, western blot analysis of cell wall proteins of the atpme3-1 line showed that the band at 33.3 kDa assigned to AtPME3 was not detected. The absence of the protein in the atpme3-1 mutant led to changes in the degree of methylesterification of galacturonic acids, without major changes in others cell wall component. For our experiments, we used the same atpme3-1 line described by Guénin et al. (2011), conserved and multiplied in our own pme mutant collection.
For the figure 3, legend of figure 3 (in the revised manuscript) was improved: white arrows indicate the localization of homogalactoronan pattern modification at the host-parasite interface and the outlines of the host-parasite interface were manually drawn (white dashed lines).
As suggested by the reviewer 4, we added figure with LM19 and LM20 signal alone, and we put figure 3 horizontally.

Reviewer 2 Report
Comments and Suggestions for Authors
The manuscript by Grandjean and colleagues demonstrates the important role of PME3 for the host plant colonization by a parasite plant Phelipanche ramosa using an elaborated system of mini-rhizotrons for monitoring these interactions. The manuscript is well-structured, easy to read and understand. It contains a big massive of results obtained using wide specter of methods that are thoroughly described in M&M section.
However, I have several points to address:
- it was shown that Atpme3-1 mutant is characterized by the reduced PME activity in roots. However, the degree of pectin methylation is also decreased there. Why PME activity reduction leads to decrease in pectin methylation?
- why authors do not discuss the activity (or changes in activity) of the Phelipanche ramose pectin-modifying enzymes and their different activity towards host plant in case of WT or mutant plant?
- in a discussion section authors should also mention events that take place in haustoria of a parasite plant, upregulation and downregulation of the PRE, and in this context a paper by Leso et al (10.1093/plphys/kiad343) which reports modification of pectin methylesterification status in the haustoria of the parasitic plant Phtheirospermum japonicum should be referred to.
- plant cell wall homeostasis is regulated by brassinosteroid feedback among other mechanisms. It should be also discussed in a context of plant-plant interactions.
Minor points:
Section 2.3. – authors should indicate the specificity of LM19 and LM20 antibodies when first mentioning them (in addition to indicating it in Figure legend and in M&M section)
L 136, 279 – should be WT instead of WS
L 342 – should be Atpme3-1 instead of Atpmei3-1
The resolution of all figures should be improved as now most of the graphs contain blurred dots and bars in Fig 3 are hardly readable.
Figure 3 should probably be rotated.
Comments on the Quality of English LanguageEnglish should be moderately corrected, with a special attention to word usage in some phrases.
Author Response

(The authors gave the same response as above.)

Reviewer 3 Report
Comments and Suggestions for Authors
It is clear from this paper that PME3 does influence the interaction between the parasite and the host. However, the results presented don’t really tell us how.
I have some minor suggestions and questions.
I think the figure legend for figure 4 needs more explanation. What are the time points for A, B, and C? You mention four time points.
Shouldn’t we know the expression levels of PME3 in the wild type, and perhaps to be sure, in the pme3 mutant?
On line 342 did you really mean Atpmei3-1?
I think you should keep the references consistent. Sometimes you use numbers and other times you use name and date. This makes finding the reference in the list inconvenient.
What are the dashed lines around the outside of the infested roots?
I’m afraid I don’t find the images in figure 3 very clear. Perhaps some arrows labelingcertain features would help.
Comments on the Quality of English LanguageStill a little French. Used et instead of and once and some non-standard arrangements of words.
Author Response

(The authors gave the same response as above.)

Reviewer 4 Report
Comments and Suggestions for Authors
The manuscript plants-3121017 is a clear study about pectin modifications and their significance in a case of parasitism. My suggestions for improvement are the following 3:
1. Please do not use so many abbreviations! The text becomes extremely hard to follow, so please keep abbreviations to a minimum and try to full-spell all the rest.
2. Because materials and methods come after results, please give some information, such as the stages of parasitism, in the results section, otherwise a reader cannot understand. Especially the light microscopy figure with these stages should be the first.
3. Because calcofluor white staining is too bright and may cover othet signals, please provide figures with LM19 and LM20 signal alone, besides the double staining. Also make an effort to put figure 3 horizontally.
Also please check for uniformity in scientific nomenclature. Some further comments/corrections/suggestions are annotated on the attached PDF.

Check on the annotated PDF, there are some points that syntax is misleading. Please rephrase using short and clear sentences.
Author Response

(The authors gave the same response as above.)
